# Chemical Profile and Multicomponent Quantitative Analysis for the Quality Evaluation of Toad Venom from Different Origins

**DOI:** 10.3390/molecules24193595

**Published:** 2019-10-06

**Authors:** Yueting Cao, Jiheng Wu, Hongye Pan, Longhu Wang

**Affiliations:** College of Pharmaceutical Sciences, Zhejiang University, Hangzhou 310058, China; caoyueting201314@zju.edu.cn (Y.C.); Wujiheng@zju.edu.cn (J.W.); 11819004@zju.edu.cn (H.P.)

**Keywords:** Chansu, HPLC-ESI-Q-TOF-MS/MS, HPLC fingerprint, quality evaluation, chemometrics

## Abstract

Toad venom (Chansu), a traditional Chinese medicine (TCM), has been widely used for treating various cancer. However, it is considerably difficult to evaluate the quality of Chansu due to its complex chemical compositions. Hence, finding the characteristic ingredients and developing a scientific and comprehensive quality evaluation method are essential for guaranteeing the safety and efficacy of Chansu. In this paper, the chemical composition database of Chansu was successfully established and HPLC-ESI-Q-TOF-MS/MS was applied for chemical profiling of the ingredients in Chansu. In total, 157 compounds were identified, including 22 amino acids, 8 alkaloids, 54 bufogenins, 63 bufotoxins, and 10 other compounds. Furthermore, HPLC fingerprints and quantitative analysis of its multicomponent were successfully developed to evaluate the quality consistency of Chansu from different origins. The results suggested that the HPLC fingerprint of Chansu could be divided into an amino acid and alkaloid region, as well as a bufogenins and bufotoxins region. The fingerprint profile of Chansu from different geographical origins were different, indicating that its quality was affected by the geographical factors. In addition, seven characteristic peaks were selected as the quantitative markers to evaluate the quality of the Chansu. The Kruskal–Wallis test illustrated that the contents of seven bufogenins in Chansu were significantly (*p* < 0.01) different among different origins. The total contents of the seven compounds ranged from 100.40 to 169.22 mg/g in 20 batches of Chansu samples. This study demonstrated that integrating HPLC-ESI-Q-TOF-MS/MS, HPLC fingerprints, and multicomponent quantitative analysis coupled with chemometrics was a comprehensive and reliable strategy for evaluation of Chansu in both qualitative and quantitative aspects. In addition, our study represented the most comprehensive characterization on the chemical compositions of Chansu, which could provide important reference information for the discovery of potential bioactive compounds.

## 1. Introduction

Chansu, the dried products of toxic secretions of *Bufo bufo gargarizans* Cantor or *Bufo melanostictus* Schneider, has been used as a traditional Chinese medicine (TCM) for thousands of years [1]. It contains a wide range of physiologically active molecules of different chemotypes, such as amino acids, alkaloids, bufogenins, bufotoxins, and so on [2,3,4]. The recent pharmacological studies have demonstrated that gamabufotalin (GB), telocinobufagin (TBG), bufotalin (BFL), cinobufotalin (CFL), bufalin (BL), cinobufagin (CBG), resibufogenin (RBG), and many other bufogenins have prominent pharmacological effects, such as cardiotonic, topical anesthetic, anti-inflammatory, immunomodulatory, and antitumor effects [5,6,7]. However, some studies indicated that many bioactive compositions in Chansu have strong toxicity [8,9]. Thus, it is essential to develop a scientific and comprehensive quality evaluation method for guaranteeing the safety and efficacy of Chansu in clinical applications.

With the continuous development of chromatography technology, the quality control of Chansu has made some progress [10,11]. But, there are still considerable challenges to evaluate the quality consistency of Chansu due to ambiguous chemical compositions. In addition, commercial Chansu was mainly processed by blending toxic secretions of toads obtained from multiple geographical origins, which may lead to the quality inconsistency among batches. Therefore, in order to scientifically evaluate and control the quality of Chansu, it is important to carry out the comprehensive characterization of chemical compositions in Chansu and explore the quality differences of the Chansu from different geographical origins.

Currently, chromatographic fingerprint has been gradually recognized as an effective approach to assess the chemical similarity of TCM by many regulatory agencies because it conformed the holistic and complicated characteristics of TCM [12,13,14]. However, fingerprint analysis can only indicate qualitative similarity, while ignoring quantitative assessments. Thus, fingerprint analysis cannot always guarantee accuracy of quality evaluation. Fortunately, multi-component quantitative analysis has been developed as a reliable method to control the quality of TCM [15,16,17]. In addition, HPLC-ESI-Q-TOF-MS/MS and HPLC fingerprint combined with chemometrics as an effective, reliable and comprehensive method were also successfully applied for quality consistency assessment of TCM [18,19,20]. Therefore, the simultaneous qualitative and quantitative of TCM can be achieved by the combination of HPLC-ESI-Q-TOF-MS/MS, HPLC fingerprint and multicomponent quantitative analysis.

TCM has the characteristic of integrity and diversity, so two or three single indicators cannot reflect their quality well. The aim of this study was to establish a comprehensive and effective quality evaluation method to evaluate the quality differences of Chansu from different origins. In this study, the comprehensive characterization of the chemical compositions in methanol extract of Chansu was accomplished by HPLC-ESI-Q-TOF-MS/MS. In addition, the HPLC fingerprint of Chansu was constructed by analyzing 20 batches of Chansu samples. Moreover, similarity analyses were utilized to assess the chemical similarity of 20 samples. The discrimination and classification of Chansu from different origins was achieved based on chemometrics. The multicomponent quantitative analysis was applied to simultaneously determine the contents of seven main compounds in Chansu. To the best of our knowledge, this is the first study to evaluate the quality differences of Chansu from different origins by HPLC-ESI-Q-TOF-MS/MS, HPLC fingerprints, and multicomponent quantitative analysis combined with chemometrics.

## 2. Results and Discussion

### 2.1. Optimization of Chromatographic Conditions for HPLC Fingerprint

Different HPLC analytical columns, mobile phase compositions, flow rates, column temperatures, detection wavelengths, and gradient elution procedures were tested to find the optimal chromatographic conditions. The optimization was done based on information amount, the signal size, the resolution of chromatographic peak, and the retention time. Four different types of columns, including XBridge^®^ BEH Amide C_18_ column (4.6 × 250 mm, 5 μm), XSelect^®^ HSS T3 C_18_ column (4.6 × 250 mm, 5 μm), ZORBAX SB C_18_ column (4.6 × 250 mm, 5 μm), and XBridge^®^ Shield RP C_18_ (4.6 × 250 mm, 5 μm) were investigated. The results showed that, compared with other columns, the XBridge^®^ Shield RP C_18_ column (4.6 × 250 mm, 5 μm) reflected better chromatographic separation effect to the target compounds. Various mobile phases (methanol–water and acetonitrile–water) containing different concentrations (0.1%, 0.2%, and 0.3%) of acid (formic acid and acetic acid) and molar concentration (10, 15, and 20 mmol) of buffer salt (ammonium formate and ammonium acetate) were compared. The results indicating that 0.3% acetic acid with 10 mmol ammonium acetate water (A) and acetonitrile (B) can effectively extend the retention time of polar alkaloids and generate the accepted peak shape. In addition, different flow rates (0.7, 0.8, 0.9, and 1.0 mL/min) were tested. Among these, 0.7 mL/min flow rate expressed higher peak numbers with better peak resolution. Different column temperatures (30, 35, and 40 °C) were also optimized. The temperature of 40 °C was selected in order to shorten the retention time. Meanwhile, the full wavelength scanning was conducted at the wavelength range of 190–400 nm by a diode array detector (DAD). All types of target compounds had strong absorption at 296 nm.

### 2.2. Method Validation for HPLC Fingerprint Analysis

The sixteen common peaks, whose sum of peak area accounted for more than 90% of all chromatographic peak area, were chosen as characteristic peaks for method validation of the HPLC fingerprint analysis. The peak 16 was selected as the reference peak because of its suitable retention time and the large peak area. Precision, repeatability, and stability were evaluated based on relative standard deviations (RSDs) of the relative retention times (RRTs) and relative peak areas (RPAs) of the sixteen characteristic peaks and the reference peak 16. The RSD values of all the RRTs and RPAs were less than 3%, which proved that the established HPLC fingerprint method was stable and reliable.

### 2.3. Method Validation for Multicomponent Quantitative Analysis of HPLC-DAD

As shown in Table 1, the multicomponent quantitative analysis method of HPLC-DAD was validated. All the seven analytes had good linearities (*R^2^* > 0.9994) within the test ranges. The limit of detection (LOD) and limit of quantitation (LOQ) for each compound in this experiment were 0.0849–0.2805 μg/mL and 0.2548–0.8416 μg/mL, respectively. For the seven analytes, the RSD values of the intraday and interday precisions varied from 0.37% to 2.36% and 1.19% to 2.92%, respectively, and the RSD values of repeatability were below 2.70%. The mean recoveries also showed a good range of 98.18% to 101.18% with RSD less than 2.11%. In addition, the RSD values of stability were less than 2.54%.

### 2.4. HPLC-ESI-Q-TOF-MS/MS Analysis for Chansu

#### 2.4.1. Establishment of Chemical Compositions Database for Chansu

The known compounds in Chansu were gathered by comprehensively searching databases, such as Pubmed, Reaxys, SciFinder, CNKI, Google Scholar, and Web of Science. The name, molecular formula, accurate molecular mass, fragment ion, and chemical structure of all the components were summarized in Microsoft Office Excel 2016 to establish a comprehensive chemical components database of Chansu.

#### 2.4.2. Identification of Chemical Constituents in Methanol Extract of Chansu

In the HPLC-ESI-Q-TOF-MS/MS analysis, the HR-MS^n^ data were imported into the PeakView Software to obtain the information of retention time, accurate molecular mass, molecular formula, and MS/MS fragment ion of compounds. Then, the chemical structures were identified based on the possible fragmentation pathway provided by the software. Finally, these results were further validated by comparison with reference standards or the reported literature data. In addition, molecular polarities were also used for assistance of identification of the components, especially the isomers with similar fragmentation pathways.

The total ion chromatograms (TICs) of the methanol extracts of Chansu in positive ion mode are shown in Figure 1. The extracted ion chromatograms (XICs) of the 22 main compounds are shown in Figure 2. In total, 157 compounds were identified or tentatively identified, including 22 amino acids, eight alkaloids, 54 bufogenins, 63 bufotoxins, and 10 other compounds (Appendix A). Among them, 10 compounds were confirmed by reference substances and 11 compounds were found in Chansu for the first time. The chemical structures of all the compounds were shown in Figure 3.

The prominent amino acids in Chansu were arginine and diacid-conjugated arginine. The diacid-conjugated arginine mainly contained C4–C11 diacids, whose names are succinyl (*n* = 2), glutaryl (*n* = 3), adipyl (*n* = 4), pimeloyl (*n* = 5), suberoyl (*n* = 6), azelayl (*n* = 7), sebacyl (*n* = 8), undecanedioyl (*n* = 9) arginine, respectively. The MS/MS fragment of compounds **2**, **10**, **11**, **15**, **22**, **31**, **37**, and **42** at *m/z* 175.1177–175.1192 indicated the loss of diacid and the retention of arginine fragments (compound **1**). Specifically, compound **22** was detected with [M + H]^+^ ion at *m/z* 331.1965 (C_14_H_27_O_5_N_4_). Its fragment ion at *m/z* 296.1593, 278.1485, 272.1481, 250.1532, 175.1179, 158.0917, 112.0870, and 70.0674 were detected. Thus, compound **22** was identified as suberoyl arginine by analyzing possible fragmentation pathway and referring to reported literature [21].

The main alkaloids in Chansu were indole alkaloids, including serotonin (5-HT) and its derivatives. Indole alkaloids found in Chansu can be conjugated with a hydroxy or sulfate group at position 5 of the indole ring. The compounds **13**, **14**, **18**, and **16** showed that the protonation in the amine nitrogen, followed by α-cleavage, led to the loss of alkylamine side chain to generate a common ion at *m/z* 160.0757 (C_10_H_10_NO^+^). Thus, the compounds **13**, **14**, **18**, and **16** were identified as 5-HT, N′-methyl-serotonin, bufotenine, and bufotenidine, respectively. Among them, the compound **13** was unambiguously identified as 5-HT by the reference substance. Their fragmentation pathways were shown in Figure 4.

Bufadienolides were considered as the major active compounds in Chansu. It mainly contains two distinct classes: One is free-form bufadienolides (bufogenins) and another is amino acid-conjugated bufadienolides (bufotoxins). Bufogenins are a type of steroids with a characteristic α-pyrone ring at C_17_. There are two kinds of classic bufogenins according to the 14-hydroxyl group or the 14,15-epoxy group. One of the most typical characteristics of bufogenins fragmentation was continuous losses of H_2_O due to multiple hydroxyl substituents in the side chain. In addition, some fragment ions resulted from the elimination of CO, HOAc, or HCHO owing to the bufogenins with 19-formyl group, 16-acetoxyl group, or 19-hydroxyl group. In our present study, a total of 54 bufogenins were identified, and nine of them, including compounds **47**, **54**, **71**, **110**, **121**, **132**, **140**, **153**, and **155**, were confirmed as Ψ-bufarenogin (Ψ-BAG), GB, arenobufagin (ABG), TBG, BFL, CFL, BL, CBG, and RBG, respectively, by standard substances. To be specific, compound **140** was eluted at t_R_ = 73.524 min with a significant [M + H]^+^ ion at *m*/*z* 387.2513 (C_24_H_35_O_4_), which produced fragment ions at *m*/*z* 369.2414 ([M + H − H_2_O]^+^) and 351.2305 ([M + H − 2H_2_O]). In addition, a conspicuous MS^2^ fragment ion was observed at *m/z* 255.2096, originating from the loss of 2H_2_O followed by elimination of the α-pyrone ring at C_17_. The t_R_ and HR-MS^n^ data of compound **140** were the same as those of standard substance bufalin. Thus, compound **140** was confirmed as bufalin. The fragmentation pathway of bufalin was shown in Figure 5A.

The bufotoxins were formed by the condensation of corresponding bufogenins and an oxalic acid, a sulfate acid, or an amino acid linked to a dicarboxylic acid. Among them, bufogenins-3-suberoyl arginine esters were the most common types. In general, bufotoxins were relatively polar compounds due to the presence of these polar moiety at the C-3 position. In our study, a total of 63 bufotoxins were identified. Among them, ten new bufotoxins were tentatively inferred by analyzing and summarizing the fragmentation rules of bufotoxins structures. The accurate mass of the bufotoxins with argininyl diacid was equal to the molecular weight of the corresponding bufogenins plus argininyl diacid minus 18 (H_2_O). For instance, the compound **137** detected [M + H]^+^ ion at *m*/*z* 699.4302 (C_38_H_59_O_8_N_4_) and displayed a characteristic fragment ion at *m*/*z* 331.1963, which was consistent with protonated suberoyl arginine (C_14_H_27_O_5_N_4_). Further, the loss of 368 fragment demonstrated that the molecular mass of relevant bufogenins was 386 (368 + 18). Thus, the compound **137** was preliminarily identified as 3-(*N*-suberoyl argininyl) bufalin (bufalitoxin). Further, the obvious fragment ion at *m*/*z* 681.4192 ([M + H − H_2_O]^+^) was noticed in its MS/MS spectrum, which was consistent with those of the published literatures [21,22]. These results confirmed compound **137** was bufalitoxin. The possible fragmentation pathway of bufalitoxin is shown in Figure 5B.

Similarly, the other bufogenins and bufotoxins were identified based on their retention time, accurate molecular mass, molecular formula, and MS/MS fragment ion. The PeakView software was used for the determination of possible fragmentation pathway of the compounds and the identification of the chemical structures. Then, these results were further validated by comparison with reference standards or the reported literatures data (Appendix A).

Particularly, the extracted ion chromatograms (Figure 2.) showed that a series of isomers were contained in Chansu simultaneously, which brings a huge challenge for the identification of these compounds. To solve this problem, the MS/MS fragments, retention times, molecular polarities differences, the reported literatures, and standard substances were used for the tentative identification of different isomers. For instance, two major isomers (compounds **54** and **110**) were detected at *m/z* 403.2467 [M + H]^+^ (C_24_H_35_O_5_) at different retention times. The t_R_ and HR-MS^n^ data of compounds **54** and **110** were the same as those of standard substances gamabufotalin and telocinobufagin, respectively. Thus, compounds **54** and **110** were confirmed as gamabufotalin and telocinobufagin, respectively. Two major isomers (compounds **117** and **133**) were detected [M + H]^+^ ion at *m/z* 401.2311 (C_24_H_33_O_5_) at different retention times. They showed similar MS/MS fragmentation patterns with a series of characteristic MS/MS ion peaks at *m/z* 383.2206, 365.2101, and 347.1997, which indicated continuous the losses of H_2_O due to hydroxyl substituents in the side chain. These MS/MS fragments information matched with the fragmentation pathway of marinobufagin and desacetylcinobufagin. The positions of hydroxyl groups in these compounds can affect their polarities. To further identify these isomers, we compared the polarities of the compounds and inferred the compounds by the relative retention time. To be specific, marinobufagin with a 5-hydroxyl group is more polar than desacetylcinobufagin with a 16-hydroxyl group (the sequence of moiety polarities were proposed as: 5-OH > 16-OH). Therefore, the compounds **117** and **133** were tentatively identified as marinobufagin and desacetylcinobufagin, respectively. Similarly, other isomers were tentatively identified based on the information of MS/MS fragments, retention times, molecular polarities differences, the reported literatures, and standard substances of compounds coupled with their possible fragmentation pathway.

### 2.5. HPLC Fingerprint and Chemical Similarity Analysis of Chansu

The HPLC fingerprints of Chansu that originated from five geographical origins were established under the optimized chromatographic conditions. The reference fingerprint was generated automatically by the median method based on the chromatographic information of the 20 batches of Chansu samples (Figure 6). A total of 33 common peaks were observed in the established fingerprints and their chemical structures were identified based on HPLC-ESI-Q-TOF-MS/MS. The HPLC fingerprints of Chansu were mainly divided into two fingerprint regions according to the relative retention time of the chromatographic peaks. The first fingerprint region was mainly composed of amino acids and alkaloids because they are relatively polar molecules, while the second fingerprint region was mainly comprised of bufogenins and bufotoxins. In particular, 14 characteristic peaks, including serotonin (P1), bufotenidine (P2), bufotenine (P3), Ψ-bufarenogin (P10), gamabufotalin (P12), arenobufagin (P16), hellebrigenin (P17), 12-β-hydroxylbufalin (P18), telocinobufagin (P23), bufotalin (P25), cinobufotalin (P28), bufalin (P30), cinobufagin (P31), and resibufogenin (P32) were selected as the representative compounds to identify the quality differences of Chansu.

Apparently, the 20 batches of Chansu from different geographical origins had similar chemical compositions. However, the relative intensity of common peaks varied dramatically in fingerprints among samples. The chemical similarity between the holistic fingerprints of samples and the reference fingerprint was analyzed to evaluate the samples’ similarity level. The results of similarity values (SVs) are shown in Appendix A. The results showed that the samples from the same origins had a similar proportion of components (0.933 ≤ SV ≤ 1.000). However, the samples from different origins were significantly different in their concentrations of components (0.625 ≤ SV ≤ 0.993). These results indicated that geographical origin was a vital factor influencing the quality consistency of Chansu.

### 2.6. The Origin Characteristics Analysis of the Chansu from Different Geographical Origins

In order to further discriminate and classify Chansu samples from different geographical origins, the 14 characteristic peak areas data of 20 samples in HPLC fingerprint was imported into SIMCA 14.0 software to perform principal component analysis (PCA) and orthogonal partial least-squares discriminant analysis (OPLS-DA). As shown, the PCA scores plot (Figure 7A) showed the initial separation among samples from different provinces, except for Jiangxi and Zhejiang provinces. The OPLS-DA scores plot (Figure 7B) was partitioned as five conspicuous regions according to all the five provinces with R^2^Y = 0.9868 and Q^2^ = 0.9428, meaning that the OPLS-DA model had a better classification and prediction ability than the PCA model. These results indicated that the established OPLS-DA model could be used as an effective tool to identify and distinguish the geographical origins of Chansu.

Although the clear separation of samples from five different geographical origins was achieved by PCA and OPLS-DA, the relative contents of the main compounds in Chansu were still indistinct. Hierarchical clustering heatmap (HCH) is a powerful tool to reflect the quantity differences by displaying the color depth in a two-dimension region, which could visualize content differences of target compounds in different samples and cluster the samples at the same time. Using Euclidean distance for similarity metrics, hierarchical clustering was performed by average linkage. The heatmap represented the size of 14 characteristic peak areas from minimum (dark blue) to maximum (red) values. The clustering result was consistent with those of PCA and OPLS-DA. The HCH divided 20 batches of samples into five categories according to five origins (Figure 8). The distance between Anhui and Henan samples was relatively close, and the distance between Jiangxi and Zhejiang samples was relatively close, illustrating that the quality of the samples from these origins was similar. However, compared with samples from the other four origins, the Sichuan samples had a relatively high P16 and a relatively low P32. As known, the accumulation of active ingredients in medicinal animals are often affected by various factors, such as geographical conditions and climatic factors. The climate differences in different regions may affect the accumulation of compositions and cause differences in the contents of these active ingredients.

Bufogenins are the major active ingredients in Chansu. The different bufogenins are functionally selective in medicine efficacy [23]. Therefore, the Chansu from different origins may have significant differences in efficacy due to the different proportions of the main bufogenins. How the differences in quality of Chansu can cause the differences in medicine efficacy needs to be further studied.

### 2.7. Quantitative Analysis of Seven Components in Chansu by HPLC-DAD

Bufogenins were considered as the main anti-tumor active ingredients in Chansu. Seven major bufogenins were selected as quantitative indicators for evaluating the quality consistency of Chansu, including GB, TBG, BFL, CFL, BL, CBG, and RBG. Generally, they accounted for more than 80% of the total content of bufogenins. Based on the established multicomponent quantitative analysis method of HPLC-DAD, the contents of seven bioactive compounds were determined simultaneously. The results are shown in Table 2. The considerable variation was found in the contents of all seven compounds (Table 2). The Kruskal–Wallis test also indicated the samples from the different origins had significant differences (*p* < 0.01).

Out of twenty samples, CBG was the most abundant active component, except for Sichuan samples (SC1-SC4). The total contents of the seven bufogenins ranged from 100.40 to 169.22 mg/g. These results demonstrated that the quality of Chansu from different origins was significantly different, which explained the reasons for the unstable quality and inconsistent therapeutic effects of the Chansu.

## 3. Materials and Methods

### 3.1. Materials and Chemicals

*Bufo bufo gargarizans* Cantor is a main species of medicinal toad, whose white secretion is raw material of Chansu. In order to evaluate the quality differences of Chansu from different origins, we collected 20 batches of fresh secretions from five representative geographical origins. The adult toads were collected in June 2018. A total of twenty batches of the *Bufo bufo gargarizans* Cantor secretions were sampled from Anhui, Henan, Jiangxi, Sichuan, and Zhejiang provinces in China. The sample information is provided in Table 3.

It is worth noting that the skin and parotid glands of toads were scraped moderately by a copper scraper. The milk-like fresh secretions obtained were placed into a clean glass dish. All collected samples were placed in coolers and transported to the laboratory immediately. Then, we prepared the Chansu by a unified processing method: By drying the fresh secretions in an oven at 60 °C for 24 h. All the dark brown, homemade Chansu obtained was crushed to powder by a mortar and passed through a 60 mesh sieve. The voucher specimens were deposited in the Modern Chinese Medicine Institute of Zhejiang University.

The reference substances, including 5-HT, Ψ-BAG, GB, ABG, TBG, BFL, CFL, BL, CBG, and RBG were purchased from Shanghai Yuanye Bio-Technology Co., Ltd. (Shanghai, China). The purities of all the compounds were determined to be higher than 98% by HPLC-DAD (1200 Series, Agilent Technologies, Tokyo, Japan). HPLC-grade acetonitrile and methanol were purchased from J&K Scientific (Beijing, China). HPLC-grade formic acid, acetic acid, ammonium formate, and ammonium acetate were obtained from Aladdin Bio-Chem Technology Co., Ltd. (Shanghai, China). Ultrapure water was generated by Milli-Q water purification system (Hangzhou, China). Other reagents were analytical-grade.

### 3.2. Preparation of Sample Solutions and Standard Solutions

The dried sample powder (60 mg) was accurately weighted and refluxed for 1 h with 100% methanol (20 mL). The extraction solutions were cooled at room temperature. Then, the lost weight of the solutions were made up for with methanol. The sample solutions were centrifuged at 4000 r/min for 5 min and the supernatant was filtered through 0.45 μm filter membrane before HPLC analysis.

The individual standard stock solutions of the 10 reference substances were prepared by dissolving reference substances in methanol (0.05 mg/mL). All the standard solutions were stored in a refrigerator at 4 °C and protected from light. The solutions were filtered through 0.45 μm filter membrane before analysis.

### 3.3. Instrumentation and Chromatographic Conditions

An Agilent 1200 system (Agilent Technologies, Tokyo, Japan) equipped with a binary gradient pump (G1312B), an on-line degasser (G1379B), an auto-sampler (G1367C), a column temperature controller (G1316B), a diode array detector (G1315C), and Agilent ChemStation was used for the HPLC fingerprint analyses and multicomponent quantitative analysis. A XBridge^®^ Shield RP C_18_ column (4.6 × 250 mm, 5 μm) was used for compound separations. HPLC fingerprint conditions were optimized as follows: The mobile phases were 0.3% acetic acid, 10 mmol ammonium acetate water (A), and acetonitrile (B). The gradient elution program was 0–7 min, 3–5% B; 7–11 min, maintain 5% B; 11–13 min, 5–15% B; 13–25 min, stay 15% B; 25–27 min, 15–24% B; 27–45 min, 24–28% B; 45–60 min, 28–32% B; 60–75 min, 32–50% B; 75–90 min, 50–3% B. The flow rate was 0.7 mL/min and the column temperature was maintained at 40 °C. The detection wavelength was set at 296 nm and the injection volume was 10 μL. The re-equilibration time was 10 min.

The conditions of multicomponent quantitative analysis of HPLC-DAD were established as follows: The mobile phases were 0.05% phosphoric acid (including 0.5% potassium dihydrogen phosphate), water (A), and acetonitrile (B). The gradient elution program was 0–7 min, 5–34% B; 7–11 min, 34–35% B; 11–18 min, 35–40% B; 18–24 min, 40–45% B; 24–35 min, 45–50% B. The flow rate was 1.0 mL/min. The column temperature was maintained at 30 °C and the injection volume was 10 μL. The detection wavelength was 296 nm. Each run was followed by re-equilibration of 5 min.

The chromatographic conditions of HPLC-ESI-Q-TOF-MS/MS were the same as those used for HPLC fingerprinting. After the compounds were separated by the liquid chromatography system (Waters Corp., Milford, MA, USA), the MS detection was conducted using a high-resolution AB Triple TOF 5600^plus^ Mass spectrometer in positive electrospray ionization (ESI) mode (AB SCIEX, Framingham, MA, USA).

The optimal MS conditions were as follows: Full scans mode within the range of *m/z* (mass/charge ratio) 100–1500 Da at a resolution of 30,000, the pressures of nebulizer gas (Gas 1) and heater gas (Gas 2) were set to 50 psi, the pressure of curtain gas (N_2_) was set to 35 psi, the source voltage was set to +5.5 kV, ion source temperature was 600 °C, and the collision energy was set at 10 V. For MS/MS acquisition mode, the parameters were almost the same except that the collision energy (CE) was set at 40 ± 20 V, ion release delay (IRD) at 67, ion release width (IRW) at 25. The information-dependent acquisition (IDA) mode-based auto-MS^2^ was performed on the precursor ions in a cycle of full scan (1 s). The scan range of *m/z* of precursor ion and product ion were set at 100–1500 Da and 50–1500 Da, respectively. The exact mass calibration was performed automatically before each analysis employing the automated calibration delivery system. Maximum allowed error was set to ±5 ppm. The mass spectrometry data were analyzed by PeakView software (AB SCIEX, version 1.2.0.3, Framingham, MA, USA).

### 3.4. Method Validation for HPLC Fingerprinting

Method validation of HPLC fingerprinting included precision, reproducibility and stability experiments. The precision was evaluated by determining the same test solution six times continuously. The reproducibility was assessed by preparation of six test solutions from the same batch samples. The stability was tested by analyzing the single test solution after storage at room temperature (25 ± 2 °C) for 0, 3, 6, 9, 12, 24, and 48 h.

### 3.5. Method Validation for the Quantitative Analysis of HPLC-DAD

The quantitative analysis method was validated by linearity, LOD, LOQ, intraday and interday precision, repeatability, accuracy, and stability. The mixed standard solutions of six concentration levels were analyzed in triplicates. Seven calibration curves were plotted based on linear regression analysis of the integrated peak areas (y) versus concentrations (x) of every composition in the standard solutions. The correlation coefficient (*R*^2^) was calculated. The LODs and LOQs were determined by diluting the working solutions until signal-to-noise ratios (S/N) of 3 and 10, respectively. The intraday precision was determined by successive analysis of the same sample solution six times within one day. The interday precision was measured by different analysts on three consecutive days. Repeatability was determined by preparing six identical samples in parallel. The recovery test was conducted to evaluate the accuracy of the developed method. To study stability of the sample solution, the same sample solution was analyzed after preparation for 0, 2, 4, 8, 12, and 24 h.

### 3.6. Data Analysis

The Kruskal–Wallis test was used to assess significant differences of samples from different origins by SPSS 20.0 software (IBM, Chicago, IL, USA). The level of significance for all hypothesis tests (*p*) was 0.05. In HPLC fingerprint analysis, similarity analysis of Chansu samples was carried out by the similarity evaluation system for Chromatographic Fingerprint of Traditional Chinese Medicine (version 2004A) software (Beijing, China). The characteristic peak areas data were imported into the SIMCA 14.0 software (Umetrics AB, Umeå, Sweden) to discriminate the different samples by performing PCA and OPLS-DA. In addition, the relative contents of major compounds in Chansu from different geographical origins were investigated by HCH (HemI).

## 4. Conclusions

In this paper, a sensitive and reliable HPLC-ESI-Q-TOF-MS/MS method was successfully established for comprehensive characterization of the chemical compositions in methanol extract of Chansu. A total of 157 compounds, including 22 amino acids, 8alkaloids, 54 bufogenins, 63 bufotoxins, and 10 other compounds were confirmed or tentatively deduced. Among them, 11 compounds were found in Chansu for the first time. In addition, an accurate and reliable HPLC fingerprint method was developed to evaluate the chemical similarity of Chansu collected from different geographical origins. Additionally, chemometric tools like PCA, OPLS-DA, and HCH were applied for distinguishing and identifying Chansu samples from different geographical origins. Meanwhile, the multicomponent quantitative analysis method of HPLC-DAD was developed to evaluate the quality of Chansu.

This study found that the HPLC fingerprint of Chansu was divided into two regions, including the amino acids and alkaloids region, as well as the bufogenins and bufotoxins region. Considerable variability was discovered in Chansu collected from five origins. The total content of the main seven bufogenins ranged from 100.40 to 169.22 mg/g in the 20 samples. In addition, the results of origin characteristics analysis of the Chansu proved that the Chansu possess geoherbalism.

This study provided comprehensive understanding of the chemical profile of Chansu. We demonstrated that HPLC-ESI-Q-TOF-MS/MS, HPLC fingerprint, and multicomponent quantitative analysis combined with chemometrics were supplementary to each other for forming a powerful and credible method for quality control of Chansu. The developed approach can be not only instrumental in elaborating more scientific quality standards of the Chansu, but also can provide a generally applicable method for discovering the quality differences of TCM from different origins.

## Figures and Tables

**Figure 1 molecules-24-03595-f001:**
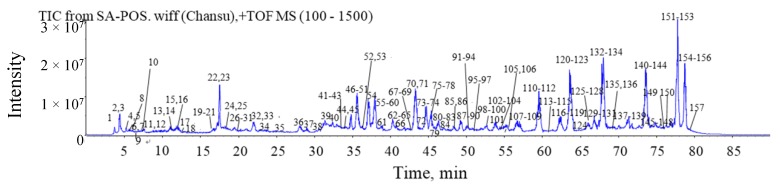
The total ion chromatograms (TICs) of Chansu methanol extracts in the positive ion mode.

**Figure 2 molecules-24-03595-f002:**
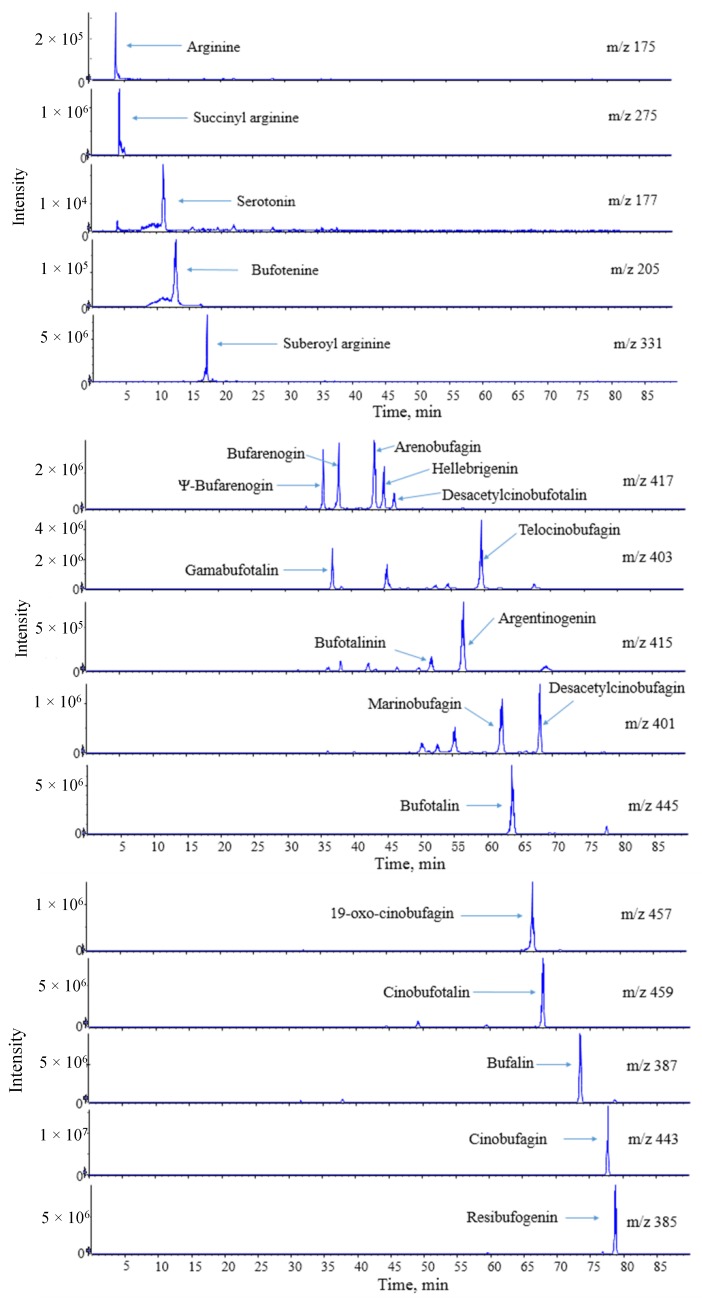
The extracted ion chromatograms (XICs) of the main compounds in the positive ion mode.

**Figure 3 molecules-24-03595-f003:**
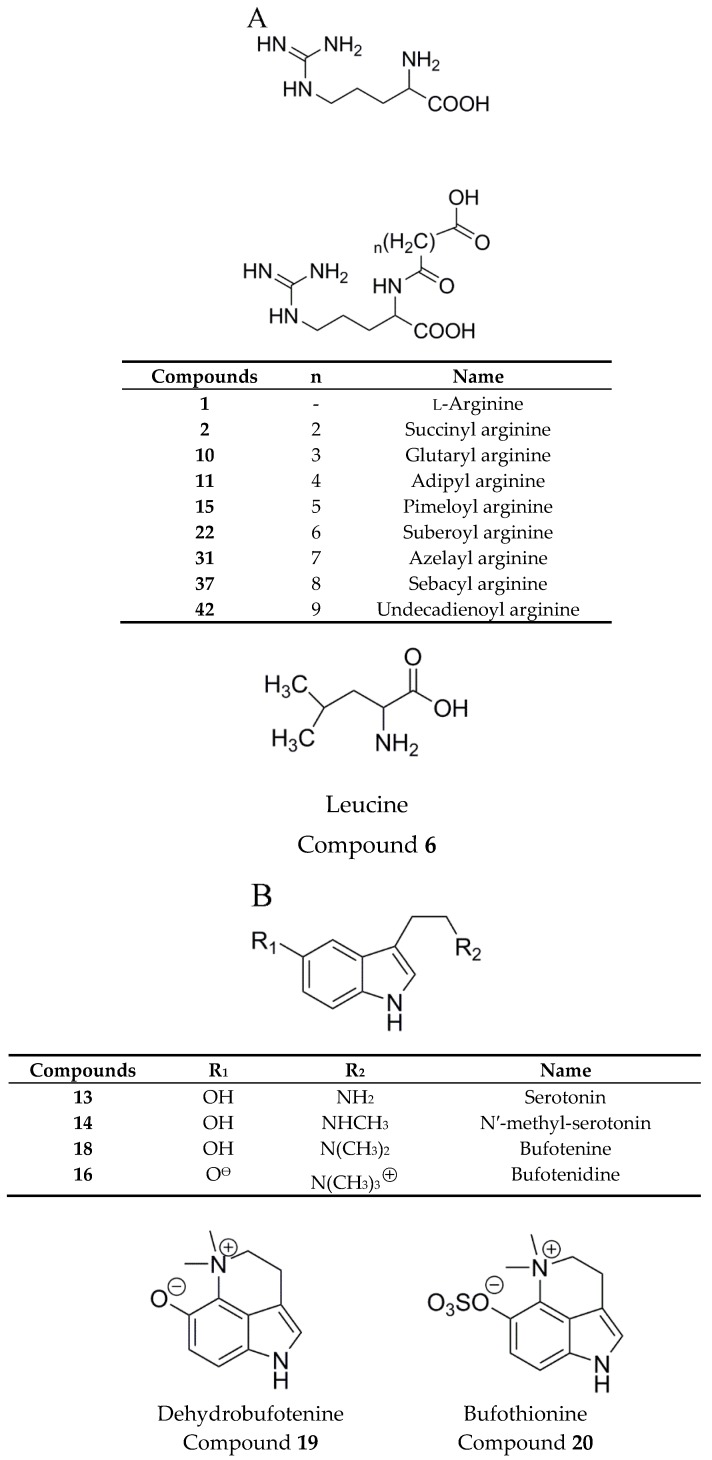
Chemical structures of compounds identified in the methanol extracts of Chansu. (**A**) amino acids; (**B**) alkaloids; (**C**) bufogenins (Group I: 14-hydroxy bufogenins; Group II: 14,15-epoxy bufogenins and Group III: Other bufogenins); (**D**) bufotoxins (Group I: 14-hydroxy bufotoxins and Group II: 14,15-epoxy bufotoxins); (**E**) other compounds.

**Figure 4 molecules-24-03595-f004:**
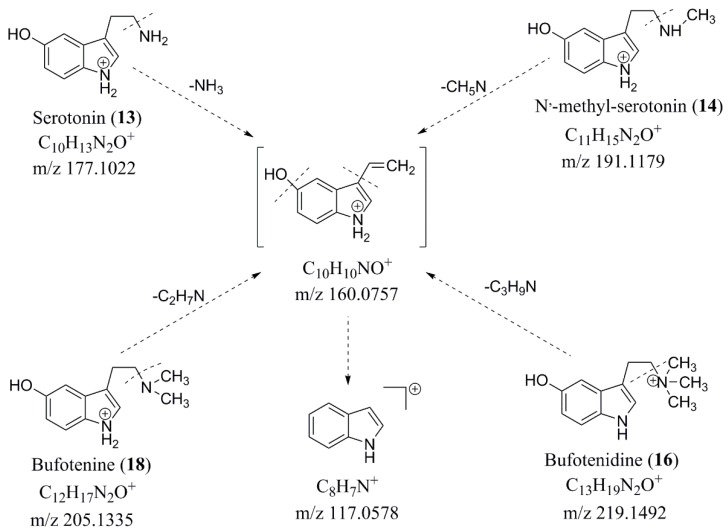
Diagnostic MS/MS fragmentation in serotonin (5-HT) and its derivatives.

**Figure 5 molecules-24-03595-f005:**
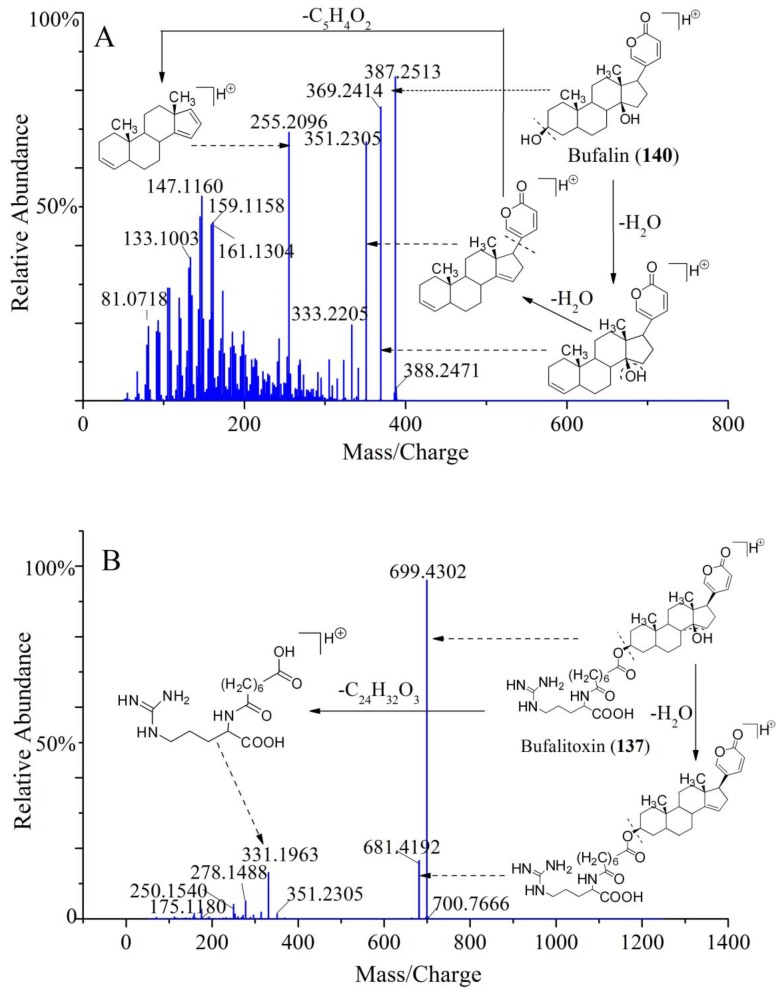
The fragmentation pathways and the typical MS/MS spectrums of bufalin (**A**) and bufalitoxin (**B**).

**Figure 6 molecules-24-03595-f006:**
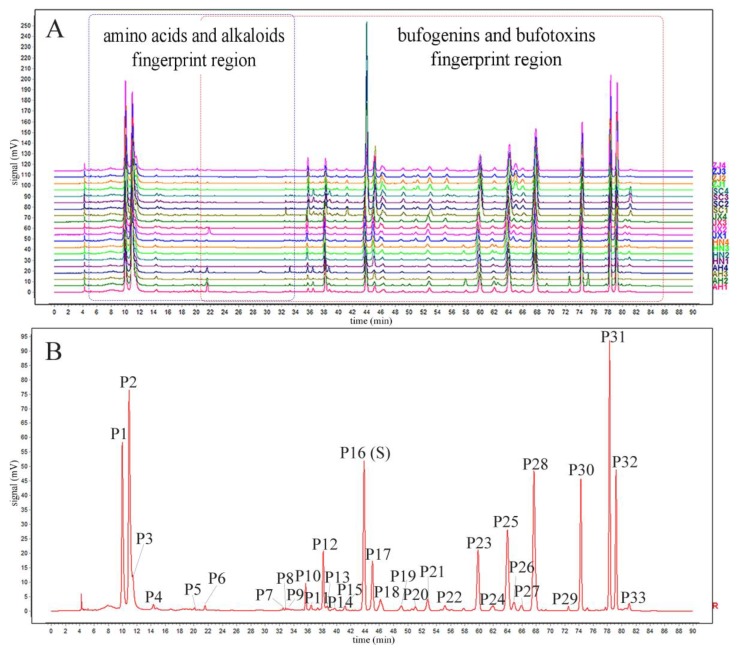
HPLC fingerprints (**A**) and reference fingerprint (**B**) of 20 Chansu samples. P1, serotonin (**13**); P2, bufotenidine (**16**); P3, bufotenine (**18**); P4, unknown; P5, unknown; P6, resibufagin stereoisomer (**32**); P7, hellebrigenol (**40**); P8, hellebrigenol-9,11-ene (**41**); P9, undecadienoyl arginine (**42**); P10, Ψ-bufarenogin (**47**); P11, bufotalinin isomer (**53**); P12, gamabufotalin (**54**); P13, gamabufotalin-3-oxalate (**55**); P14, 19-hydroxylbufalin (**60**); P15, 11α-hydroxytelocinobufagin (**65**); P16, arenobufagin (**71**, reference peak); P17, hellebrigenin (**74**); P18, 12-β-hydroxylbufalin (**78**); P19, cinobufaginol (**88**); P20, resibufaginol (**93**); P21, 19-oxo-cinobufotalin (**101**); P22, 19-oxo-bufalin (**106**); P23, telocinobufagin (**110**); P24, marinobufagin (**117**); P25, bufotalin (**121**); P26, resibufagin (**125**); P27, 19-oxo-cinobufagin (**128**); P28, cinobufotalin (**132**); P29, bufalitoxin (**137**); P30, bufalin (**140**); P31, cinobufagin (**153**); P32, resibufogenin (**155**); P33, 3-epi-cinobufagin (**157**).

**Figure 7 molecules-24-03595-f007:**
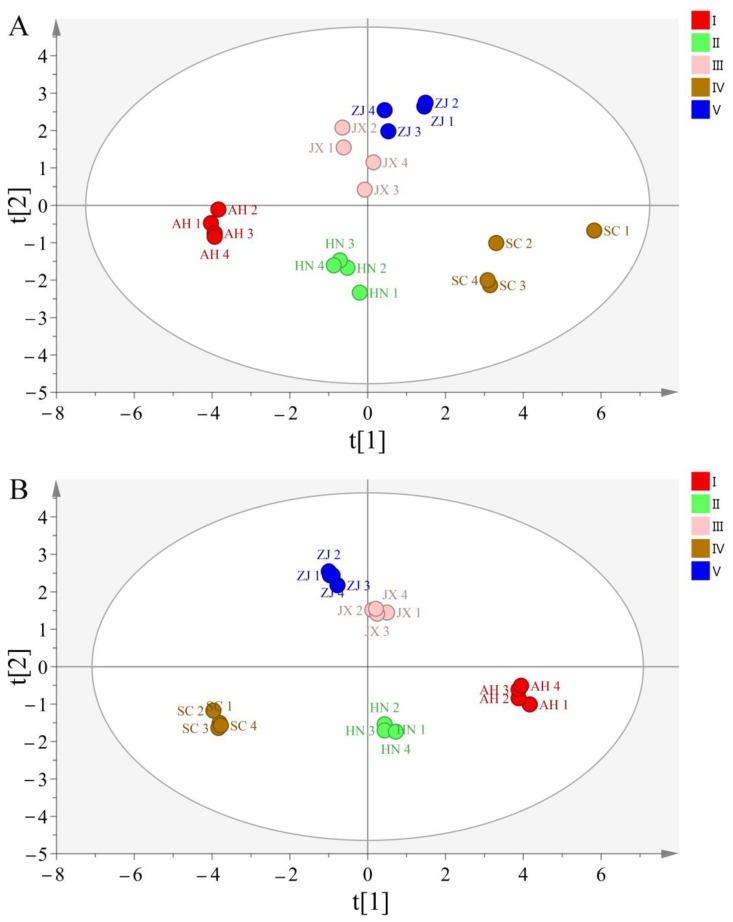
The PCA: principal component analysis (**A**) and OPLS-DA: orthogonal partial least-squares discriminant analysis (**B**) scores plot of 20 batches of Chansu samples based on peak area data of fourteen characteristic compounds.

**Figure 8 molecules-24-03595-f008:**
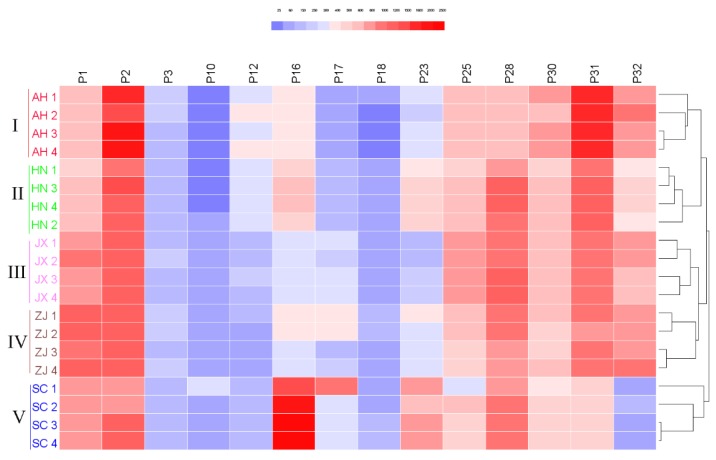
The HCH: hierarchical clustering heatmap of 20 batches of Chansu samples from five provinces based on peak areas data of fourteen characteristic compounds.

**Table 1 molecules-24-03595-t001:** Method validation for simultaneous quantification of seven compounds in Chansu. LOD: limit of detection, LOQ: limit of quantification, RSD: relative standard deviation.

Compound	Regressive Equation	Linear Range (μg/mL)	R^2^	LOD (μg/mL)	LOQ (μg/mL)	Precision		Repeatability RSD% (*n* = 6)	Recovery		Stability RSD%
						Intraday RSD% (*n* = 6)	Interday RSD% (*n* = 6)		Mean	RSD% (*n* = 6)	
**GB**	y = 8.5450x + 0.4726	2.632–263.2	0.9995	0.1053	0.3290	0.86	1.73	1.76	99.50	1.44	1.19
**TBG**	y = 8.4668x − 2.6368	3.380–338.0	0.9995	0.2253	0.6760	0.74	1.19	1.30	101.18	1.68	0.97
**BFL**	y = 7.6345x − 1.2852	2.548–254.8	0.9995	0.0849	0.2548	0.37	2.92	1.47	100.08	1.53	1.45
**CFL**	y = 7.8537x − 2.4578	3.852–385.2	0.9995	0.2568	0.7704	0.37	2.45	1.51	98.91	1.52	0.90
**BL**	y = 8.7099x − 2.5159	2.560–256.0	0.9995	0.1024	0.3012	0.74	1.48	1.41	99.83	2.11	1.81
**CBG**	y = 7.9887x − 3.5450	4.208–420.8	0.9994	0.2805	0.8416	0.57	2.07	1.40	98.18	1.82	0.58
**RBG**	y = 8.5660x − 2.4699	3.424–342.4	0.9995	0.2283	0.6848	2.36	2.73	2.70	100.90	1.83	2.54

**Table 2 molecules-24-03595-t002:** The contents (mg/g) of the seven bufogenins in 20 batches of Chansu samples from different geographical origins. GB: gamabufotalin, TBG: telocinobufagin, BFL: bufotalin, CFL: cinobufotalin, BL: bufalin, CBG: cinobufagin, RBG: resibufogenin.

No.	GB	TBG	BFL	CFL	BL	CBG	RBG
AH 1	10.99 ± 0.26	10.33 ± 0.16	19.11 ± 0.14	19.72 ± 0.35	27.23 ± 0.20	56.35 ± 0.62	25.50 ± 0.21
AH 2	12.80 ± 0.46	8.14 ± 0.14	18.89 ± 0.25	17.92 ± 0.45	23.27 ± 0.27	54.10 ± 0.67	28.90 ± 0.25
AH 3	11.48 ± 0.21	9.53 ± 0.04	19.07 ± 0.20	18.93 ± 0.04	25.64 ± 0.14	55.44 ± 0.45	26.99 ± 0.24
AH 4	11.56 ± 0.26	9.22 ± 0.08	18.99 ± 0.03	18.78 ± 0.07	24.28 ± 1.25	55.47 ± 0.47	27.29 ± 0.41
HN 1	9.51 ± 0.48	17.05 ± 0.69	19.33 ± 0.51	31.13 ± 0.83	17.39 ± 0.72	35.17 ± 0.74	14.78 ± 0.99
HN 2	10.48 ± 0.53	15.16 ± 0.70	19.57 ± 0.96	27.54 ± 0.85	18.77 ± 0.71	37.31 ± 1.84	12.97 ± 0.90
HN 3	9.56 ± 0.38	16.26 ± 0.78	18.88 ± 0.78	30.28 ± 1.42	18.04 ± 0.70	40.78 ± 0.56	14.17 ± 0.31
HN 4	9.72 ± 0.57	15.74 ± 0.38	18.47 ± 0.84	30.57 ± 0.46	17.99 ± 0.72	39.35 ± 1.80	15.07 ± 0.61
JX 1	6.24 ± 0.37	6.35 ± 0.36	25.19 ± 0.86	26.27 ± 0.78	20.64 ± 0.47	32.23 ± 0.55	23.72 ± 0.49
JX 2	6.26 ± 0.18	6.81 ± 0.33	24.14 ± 1.02	26.02 ± 0.59	18.94 ± 1.07	32.32 ± 1.52	23.09 ± 1.19
JX 3	8.12 ± 0.09	8.46 ± 0.26	27.23 ± 0.27	29.74 ± 0.22	18.53 ± 0.98	28.99 ± 0.49	17.38 ± 1.19
JX 4	7.17 ± 0.10	7.94 ± 0.14	25.29 ± 0.77	29.54 ± 0.48	18.87 ± 0.14	30.84 ± 0.36	18.62 ± 0.26
SC 1	7.21 ± 0.20	25.56 ± 0.65	11.11 ± 0.30	26.25 ± 0.74	12.71 ± 0.43	14.64 ± 0.42	2.92 ± 0.18
SC 2	5.89 ± 0.36	21.97 ± 0.28	19.00 ± 0.23	30.56 ± 0.39	15.43 ± 0.06	16.22 ± 0.20	6.71 ± 0.16
SC 3	6.69 ± 0.43	24.57 ± 0.51	18.05 ± 0.48	31.35 ± 0.56	16.36 ± 0.39	16.06 ± 0.37	2.91 ± 0.19
SC 4	6.58 ± 0.29	24.23 ± 0.38	17.74 ± 0.25	31.39 ± 0.66	16.13 ± 0.27	15.81 ± 0.06	2.90 ± 0.13
ZJ 1	5.34 ± 0.11	11.86 ± 0.28	19.84 ± 0.48	22.08 ± 0.84	18.68 ± 0.58	30.02 ± 0.72	23.92 ± 0.46
ZJ 2	5.10 ± 0.14	11.41 ± 0.26	19.17 ± 0.48	21.73 ± 0.50	17.97 ± 0.41	29.17 ± 0.77	23.49 ± 0.47
ZJ 3	4.28 ± 0.09	9.03 ± 0.20	16.45 ± 0.19	19.08 ± 0.68	18.07 ± 0.15	29.56 ± 0.38	26.98 ± 0.19
ZJ 4	4.54 ± 0.13	9.65 ± 0.27	17.33 ± 0.25	19.67 ± 1.07	18.91 ± 0.28	30.99 ± 0.40	28.50 ± 0.43
RSD (%)	32.46	46.66	18.06	20.07	18.30	39.08	48.85

**Table 3 molecules-24-03595-t003:** Sources information of 20 batches of Chansu samples from different geographical origins.

No.	Source	No.	Source
AH 1	Anhui	JX 3	Jiangxi
AH 2	Anhui	JX 4	Jiangxi
AH 3	Anhui	SC 1	Sichuan
AH 4	Anhui	SC 2	Sichuan
HN 1	Henan	SC 3	Sichuan
HN 2	Henan	SC 4	Sichuan
HN 3	Henan	ZJ 1	Zhejiang
HN 4	Henan	ZJ 2	Zhejiang
JX 1	Jiangxi	ZJ 3	Zhejiang
JX2	Jiangxi	ZJ4	Zhejiang

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
