# Peer review of "Chemical Profile and Multicomponent Quantitative Analysis for the Quality Evaluation of Toad Venom from Different Origins"

_molecules, 2019, doi:10.3390/molecules24193595_

Round 1

Reviewer 1 Report

Review comments
Chansu has been traditionally used for treating various disorders including cancers in many Asian countries. With its unclear chemical composition, it is really unsafe for medical uses. The authors analysed the chemical composition of Chansu from different geographical origins, and they found that the quality of the Chansu is affected by the geographical factors and the fingerprint regions vary from different geographical origins. They claimed that their study represented the most comprehensive characterization on the chemical compounds of Chansu, which provides important reference for Chansu’s potential bioactive components.

In this study, this reviewer only has two concerns about the variation among
different geographic origins of Chansu:
First, the standard processing procedure of Chansu including the venom collection was not described. All the variations also could stem from different processing procedures before the dried Chansu samples, instead of the geographical influences.
Second, there is no way to confirm those samples are all from exactly the same or different species or subspecies of toads. Nevertheless, this study could still serve as a reference for the general composition of Chansu, although more refine and indepth physiological studies are required for those specific bioactive components in Chansu.

Author Response

Response to Reviewer 1 Comments

Point 1: First, the standard processing procedure of Chansu including the venom collection was not described. All the variations also could stem from different processing procedures before the dried Chansu samples, instead of the geographical influences.

Response 1:

Thanks for reviewer’s kind comment. In the experimental design, we considered

that the commodity Chansu prepared by different processing procedures may have a certain impact on the analysis results. Therefore, we selected fresh venom as the research object. The fresh venom was obtained by directly scrape the skin and parotoid glands of toads from different geographical origins. Then, we prepared the Chansu by an unified processing method (by drying the fresh secretions in an oven at 60 degrees for 24 hours ).

This method can avoid the influence of different processing procedures when

analyzing the quality difference of Chansu among different geographical origins. This content was mentioned in the "Materials and Methods" section, line 375-376.

In addition, we added the venom collection method in revised manuscript, line

367-378.

Point 2: Second, there is no way to confirm those samples are all from exactly the same or different species or subspecies of toads. Nevertheless, this study could still serve as a reference for the general composition of Chansu, although more refine and indepth physiological studies are required for those specific bioactive components in Chansu.

Response 2:

Thank you for your recognition of my article. In order to rule out the differences

among different species, we chose the Bufo bufo gargarizans Cantor (The Chinese Pharmacopoeia stipulates that the Bufo bufo gargarizans Cantor is one of the sources of Chansu). We identified Bufo bufo gargarizans Cantor by the morphological characteristics of toads. Because the common toads, including Bufo melanostictus and Buforaddei, in China, are very different from Bufo bufo gargarizans Cantor in the appearance. The pharmacological activity of specific bioactive components in Chansu and the efficacy differences of Chansu stemmed from different toad species should be further studied.

Reviewer 2 Report

The topic is not new but, but, on a quick Internet search, this manuscript presents one of the the most complete and comprehensive approach. The current manuscript also provides novel information regarding new possible compounds.  Experiments are very structured, all analytical methods were optimised.  The statistic calculation offers a complete image on the results of the experimental work.  The content of the paper is very interesting and should be published after a major revision.

However, some aspects regarding the MS analysis should be clarified.

Observation:

Provide work resolution for MS and MS-MS analysis. R 378 – you provided MS condition without resolution but you don’t specify the MS-MS condition.  What approach did you use for MS analysis? Full scan followed by all ions fragmentation, full scan followed by targeted MS-MS or other? Please explain how (figure 2 – extracted chromatogram in full scan) you identified the compounds with the same mass 417 (and likewise 403, 415, 401) Mode explanation are needed for: R 142 ‘The known compounds were identified by referencing the information of pre-established chemical compositions database of Chansu (?? – fragments identification in a targeted MS-MS approach?) combined with the reference substances and the isomers with completely identical reference information were tentatively distinguished by comparing the polarity differences of different structures and relative retention time.’ What do you mean with ‘comparing polarity differences’? Did you use only literature references for fragment identification of the known compound (not those reference standards confirmed)? A software as Mass Frontier or other could be more useful in providing possible fragmentation pathway of the compounds and leading to more accurate presumptive identification. R222 – ‘the known bufogenins and bufotoxins were identified based on the information of mass spectrometry and literatures’. Please specify at what information regarding mass spectrometry you referred. You compared the fragments with literature or with a prediction software? What resolution was used for MS-MS analysis? What was the used working mode? ‘multicomponent quantitative analysis method was validated’ – you should specify that is a UHPLC method Why did you use UHPLC for quantification and not MS analysis? Some peaks are overlapping.  (ex 2 with 3). LC or LC-MS quantification for 7 bioactive compounds in chansu was already described in many publication (Ma, Xiao Chi, et al. "Simultaneous quantification of seven major bufadienolides in three traditional Chinese medicinal preparations of chansu by HPLC-DAD." Natural product communications2 (2009): 1934578X0900400203.; Ye, Min, and De‐an Guo. "Analysis of bufadienolides in the Chinese drug ChanSu by high‐performance liquid chromatography with atmospheric pressure chemical ionization tandem mass spectrometry." Rapid Communications in Mass Spectrometry: An International Journal Devoted to the Rapid Dissemination of Up‐to‐the‐Minute Research in Mass Spectrometry19.13 (2005): 1881-1892. Etc) In table 1S you reported confirmation with reference standards for 10 compounds. You could quantify all not just 7. Because I am not a specialist in TCM I am not sure if 20 batches are representative for chansu characterisation. Please specify.

Author Response

Response to Reviewer 2 Comments

Point 1: Provide work resolution for MS and MS-MS analysis. R378–you provided MS condition without resolution but you don’t specify the MS-MS condition. What approach did you use for MS analysis? Full scan followed by all ions fragmentation, full scan followed by targeted MS-MS or other?

Response 1:

Thanks for reviewer’s kind comment. The work resolution for MS and MS-MS

were higher than 30,000 ( high-resolution, HR-MS). For MS/MS acquisition mode, the parameters were almost the same as MS, except that the collision energy (CE) was set at 40±20 V, ion release delay (IRD) at 67, ion release width (IRW) at 25. The information dependent acquisition (IDA) mode based auto-MS2 was performed on the precursor ions in a cycle of full scan (1 s).

These parameters of the MS and MS-MS conditions had be added to the revised

manuscript, Line 425-426, Line 429-433.

Point 2: Please explain how (figure 2 - extracted chromatogram in full scan) you identified the compounds with the same mass 417 (and likewise 403, 415, 401).

Response 2:

Thanks for reviewer’s kind comment. The extracted ion chromatograms (XICs)

showed that a series of isomers contained in Chansu simultaneously, which brings a huge challenge for the identification of these compounds. To solve this problem, the MS/MS fragments, retention times, polarity differences, the reported literatures and standard substances were used for the tentatively identification of different isomers.

 Five major isomes (compounds 47, 56, 71, 74, 80) were detected [M+H]+ ion at m/z 417.2263 [M+H]+ (C24H33O6) at different retention times. Among them, the tR and HR-MSn data of compounds 47 and 71 were the same as those of standard substance Ψ-Bufarenogin and Arenobufagin, respectively. In addition, compounds 56, 74 and 80 showed different MS/MS fragmentation patterns. Their MS2 fragment ion with the strongest abundance were observed at m/z 399.2161, 335.1994 and 363.1942, respectively. Among them, compound 74 produced fragment ions at m/z 399.2148, 381.2047, 363.1944 and 335.1994, indicated the loss of 3H2O followed by elimination of CO. These MS/MS fragments information matched with the fragmentation pathway of Hellebrigenin. Analogously, the MS/MS fragments information of compounds 56 and 80 matched with the fragmentation pathway of Bufarenogin and Desacetylcinobufotalin. In addition, these MS/MS fragment ions were also consistent with those of the published literatures. Thus, compounds 47 and 71 were confirmed as Ψ-Bufarenogin and Arenobufagin. The compounds 56, 74 and 80 were tentatively dentified as Bufarenogin,Hellebrigenin and Desacetylcinobufotalin, respectively.

Two major isomers (compounds 54 and 110) were detected [M+H]+ ion at m/z 403.2467 [M+H]+ (C24H35O5) at different retention times. The tR and HR-MSn data of compounds 54 and 110 were the same as those of standard substance Gamabufotalin and Telocinobufagin, respectively. Thus, compounds 54 and 110 were confirmed as Gamabufotalin and Telocinobufagin, respectively. 

Two major isomers (compounds 96 and 107) were detected [M+H]+ ion at m/z 415.2101(C24H31O6) at different retention times. They showed different MS/MS fragmentation patterns. Their MS2 fragment ion with the strongest abundance were observed at m/z 351.1938 and 397.2008, respectively. Among them, compound 96 produced fragment ions at m/z 397.2010, 379.1895, 361.1783 and 351.1938, indicated the loss of 2H2O followed by elimination of CO. These MS/MS fragments information matched with the fragmentation pathway of Bufotalinin. Analogously, the MS/MS fragments information of compound 107 matched with the fragmentation pathway of Argentinogenin. In addition, these MS/MS fragment ions were also consistent with those of the published literatures. Thus, compounds 96 and 107 were tentatively identified as Bufotalinin and Argentinogenin, respectively.

Two major isomers (compounds 117 and 133) were detected [M+H]+ ion at m/z

401.2311 (C24H33O5) at different retention times. They showed similar MS/MS

fragmentation patterns with a series of characteristic MS/MS ion peaks at m/z

383.2206, 365.2101 and 347.1997, which indicated continuous losses of H2O due to hydroxyl substituents in the side chain. These MS/MS fragments information matched with the fragmentation pathway of Marinobufagin and Desacetylcinobufagin. The positions of hydroxyl groups in these compounds can affect their polarities. To further identify these isomers, we compared the polarities of the compounds and inferred the compounds by the relative retention time. To be specific, Marinobufagin with a 5-hydroxyl group is more polar than Desacetylcinobufagin with a 16-hydroxyl group (The sequence of moiety polarities were proposed as: 5-OH > 16-OH). Therefore, the compounds 117 and 133 were tentatively identified as Marinobufagin and Desacetylcinobufagin, respectively. 

The explanation of the representative isomers have been given in the revised manuscript, Line 245-268.

Point 3: Mode explanation are needed for: R 142 ‘The known compounds were

identified by referencing the information of pre-established chemical compositions database of Chansu (??–fragments identification in a targeted MS-MS approach?) combined with the reference substances and the isomers with completely identical reference information were tentatively distinguished by comparing the polarity differences of different structures and relative retention time.’ What do you mean with comparing polarity differences’? Did you use only literature references for fragment identification of the known compound (not those reference standards confirmed)? A software as Mass Frontier or other could be more useful in providing possible fragmentation pathway of the compounds and leading to more accurate presumptive identification.

Response 3:

Thanks for reviewer’s kind suggestion. Before HR-MS analysis, we searched a

lot of literatures on the chemical compositions of Chansu. The name, molecular

formula, accurate molecular mass, fragment ion and chemical structure of all the components were summarized. During the HR-MS analysis, the HR-MSn data were imported into the PeakView Software to obtain the information of retention time, accurate molecular mass, molecular formula and MS/MS fragment ion of compounds. Then, the chemical structures were identified based on the possible fragmentation pathway of the compounds provided by the software. Finally, these results were further validated by comparison with reference standards or the reported literature data.

Yes, for the isomers with similar fragmentation pathway, the molecular polarities

were used to differentiate them from each other. The sequence of moiety polarities were proposed as: – OH > 19-CHO > 11 (12)-CO > 14, 15-epoxy, and 12-OH >11-OH > 5-OH > 16-OH > 19-OH.

We used simultaneously literature references and reference standards for

fragment identification of the known compounds. However, because some reference standards were difficult to obtain, 10 compounds were confirmed accurately by comparison with the reference standards, and the other compounds were verified by comparison with the reported literature data.

We have rewritten these contents in the revised manuscript. Line 143-150, Line

209-218, Line 236-241.

Point 4: R222–‘the known bufogenins and bufotoxins were identified based on the information of mass spectrometry and literatures’. Please specify at what information regarding mass spectrometry you referred. You compared the fragments with literature or with a prediction software? What resolution was used for MS-MS analysis? What was the used working mode?

Response 4:

Thanks for reviewer’s kind suggestion. The information of mass spectrometry

included retention time, accurate molecular mass, molecular formula and MS/MS

fragment ion. These information have been specified in the revised manuscript. Line  236-237.

The HR-MSn data were analysed by PeakView Software (AB SCIEX, version

1.2.0.3). Firstly, the PeakView Software was used for the determination of possible fragmentation pathway of the compounds and the presumptive identification of the chemical structures. Then, these results were further validated by comparison with reference standards or the reported literature data. 

The work resolution for MS-MS analysis was higher than 30,000 

(high-resolution, HR-MS). The information dependent acquisition (IDA) mode based auto-MS2 was performed on the precursor ions in a cycle of full scan (1 s). These contents had be added to the revised manuscript. Line 425-433.

Point 5: ‘multicomponent quantitative analysis method was validated’–you should specify that is a UHPLC method. Why did you use UHPLC for quantification and not MS analysis? Some peaks are overlapping. (ex 2 with 3).

Response 5:

Thanks for reviewer’s kind suggestion. We specified that we established a

HPLC-DAD method for multi-component quantitative analysis, in revised manuscript. Line 122-124.

HPLC-DAD can be used to quantify the components with UV absorption, and

the peak purity of the target components had be detected by full scan with DAD. The purity factors of the seven peaks used for quantification were all greater than 990, which was suitable for quantitative analysis by this method. MS has high sensitivity and it is suitable for analyzing the components with low concentration and no UV absorption. Mass spectrometry has advantages in analyzing complex samples with a lot of components. However, MS quantitative analysis is costly. Therefore, in solving the multicomponent quantitative problem, the accuracy and stability of HPLC-DAD has already been satisfactory, so we don’t use more complex and expensive MSmethods to quantify these components.

Point 6: LC or LC-MS quantification for 7 bioactive compounds in chansu was

already described in many publication (Ma, Xiao Chi, et al. "Simultaneous

quantification of seven major bufadienolides in three traditional Chinese medicinal preparations of chansu by HPLC-DAD." Natural product communications2 (2009):1934578X0900400203.; Ye, Min, and De‐an Guo. "Analysis of bufadienolides in the Chinese drug ChanSu by high‐performance liquid chromatography with atmospheric pressure chemical ionization tandem mass spectrometry." Rapid Communications in Mass Spectrometry: An International Journal Devoted to the Rapid Dissemination of Up‐to‐the‐Minute Research in Mass Spectrometry19.13 (2005): 1881-1892. Etc)

In table 1S you reported confirmation with reference standards for 10 compounds. You could quantify all not just 7.

Response 6:

Thanks for reviewer’s kind comment. About “Simultaneous quantification of

seven major bufadienolides in three traditional Chinese medicinal preparations of chansu by HPLC-DAD”, this paper only developed a HPLC-DAD method to utilize for the quality control of the three traditional Chinese medicinal preparations of ChanSu. It did not conduct qualitative analysis of Chansu. About “Analysis of bufadienolides in the Chinese drug ChanSu by high‐performance liquid chromatography with atmospheric pressure chemical ionization tandem mass spectrometry.” this paper utilized atmospheric pressure chemical ionization tandem mass spectrometry to analyze the bufadienolides in Chansu. It did not perform quantitative analysis of Chansu. Our manuscript integrated HPLC-MS/MS, HPLCfingerprints and multicomponent quantitative analysis coupled with chemometrics. It was a comprehensive and reliable strategy for quality evaluation of Chansu in both qualitative and quantitative aspects.

In this manuscript, the purpose of multicomponent quantitative analysis was to

evaluate the quality of Chansu. When establishing an analysis method, it is necessary to think out the selection of appropriate quantitative components. In addition, It is very important that whether the analysis method is stable and reliable, simple and economical, and the analysis time is short enough.

We believe that in quantitative evaluation of Chansu, the more quantitative

components do not mean the better. On the one hand, it will take longer time to

analyze more components, which is not conducive to the quality assessment of

large-scale products. On the other hand, some ingredients are inactive or very low concentrations in Chansu, and their reference standards are very expensive. It is almost meaningless to quantify these components. Therefore, under the premise that the quality difference can be assessed, the selected indicators should be as little as possible, rather than more. Thus, based on the literature and our results, we selected seven active and high-content components in the Chansu as quantitative components, including gamabufotalin (GB), telocinobufagin (TBG), bufotalin (BFL), cinobufotalin (CFL), bufalin (BL), cinobufagin (CBG) and resibufogenin (RBG). Generally, they accounted

for more than 80% of the total content of bufogenins. These indicators can reflect the quality differences of Chansu from different regions. By establishing an accurate, reliable and rapid multi-component quantitative method, we hope to better evaluate the quality of Chansu.

Point 7: Because I am not a specialist in TCM. I am not sure if 20 batches are

representative for chansu characterisation. Please specify.

Response 7:

Thanks for reviewer’s kind comment. The establishment of TCM fingerprints

requires that the sample size are greater than 10 batches. In this article, we have collected 20 different batches of samples, this sample size is not small in the same type of article: (1) Lu, H.Y.; Ju, M.Z.; Chu, S.S. “Quantitative and Chemical Fingerprint Analysis for the Quality Evaluation of Platycodi Radix Collected from Various Regions in China by HPLC Coupled with Chemometrics” Molecules. 2018,23(7): 1823.

(2) Cao, X.X; You, G.J.; Li, H.H. “Comparative Investigation for Rotten Xylem (kuqin) and Strip Types (tiaoqin) of Scutellaria baicalensis Georgi Based on Fingerprinting and Chemical Pattern Recognition” Molecules. 2019, 24(13): 2431.

(3) Nzeuwa, I.B.Y.; Xia, Y.Y.; Qiao, Z. et al. “Comparison of the origin and phenolic contents of Lycium ruthenicum Murr. by high‐performance liquid chromatography fingerprinting combined with quadrupole time‐of‐flight mass spectrometry and chemometrics” Journal of separation science. 2017, 40(6): 1234-1243).

We think the results given by the existing sample size can represent the characteristics of Chansu.

Reviewer 3 Report

This paper typically uses modern analytical methods to qualitatively and quantitatively analyze the characteristic components of traditional Chinese medicine. Its purpose is to develop and set up the standard to ensure the safety, efficacy, and consistency of traditional Chinese medicines in clinical application. Taking toad venom as an example, HPLC-ESI-Q-TOF-MS/MS, HPLC fingerprints, and multicomponent quantitative analysis were combined to evaluate the quality difference of 20 batches of toad venom in 5 regions. The experiment was very rigorous that the precision, repeatability, and stability of HPLC fingerprint analysis method were verified. The characteristic ingredients of toad venom were qualitatively analyzed by HPLC-ESI-Q-TOF-MS/MS, and a database of 157 compounds was established, of which 11 were newly discovered compounds. Qualitative and quantitative analysis of samples from different regions was performed using HPLC fingerprints and multi-component quantitative analysis. Seven characteristic peaks were selected as quantitative indicators to evaluate the difference in quality of toad venom from different regions. This study has certain reference value for the standardization of other traditional Chinese medicines. There are a few minor issues as follows:

1. The characteristic peaks of Sichuan's samples are relatively different than those of other provinces (The Sichuan samples had a relatively high P16 and a relatively low P32). What do the authors think is the possible reason(s)? Is there an advantage in the difference in medicine efficacy?

2. Line 153 Figure1.The total ion chromatograms (TICs) of Chansu methanol extract in the positive ion mode. "Figure1" should add a space, such as Figure 1.

3 Line 155 Figure2.The extracted ion chromatograms (XICs) of the main compounds in the positive ion mode. "Figure2" should add a space, such as Figure 2.

4 Line 232: generated automatically by median method based onthe chromatographic information. "onthe" should add a space.  "median method" should be the median method.

Author Response

Response to Reviewer 3 Comments

Point 1: The characteristic peaks of Sichuan's samples are relatively different than those of other provinces (The Sichuan samples had a relatively high P16 and a relatively low P32). What do the authors think is the possible reason(s)? Is there an advantage in the difference in medicine efficacy?

Response 1:

Thanks for reviewer’s kind comment. First of all, in this study, the Chansu of

different origins are all derived from Bufo bufo gargarizans Cantor. Therefore, the differences of chemical compositions in Chansu from different origins could not stem from different species. Secondly, accumulation of active ingredients in medicinal animals are often affected by various factors, such as, geographical conditions and climatic factors during the growing period. The differences of active ingredients in different origins may be related with climatic conditions. Climatic factors may regulate the accumulation of compositions and cause contents variation of active ingredients.

The toads like humid environments. The precipitation for China, is significantly

different in different provinces. In particular, Sichuan province is far from other

provinces (Anhui, Henan, Jiangxi, and Zhejiang). Compared to other provinces, the Sichuan are more different in climatic conditions. In addition, the annual total precipitation has significantly decreased in Sichuan Basin (Zhai, P. M., Zhang, X. B., Wan, H. & Pan, X. H. Trends in Total Precipitation and Frequency of Daily Precipitation Extremes over China.). We think that the differences of moisture in different regions may cause the differences of compounds contents. P16 is arenobufagin and P32 is resibufogenin. They are all bufogenins, a type of steroids with a characteristic α-pyrone ring at C17. The biosynthesis mechanism of these compounds may be shifted in different environmental conditions (including precipitation differences). In addition, bufogenins may also be transformed by certain microorganisms and the differences of chemical components may result from biotransformation of the toad skin microbiome in different habits.

Bufogenins are the major active ingredients in Chansu. The different bufogenins

are functionally selective in medicine efficacy. Therefore, the Chansu from different origins may have significant differences in efficacy due to the different proportions of the main bufogenins. (For example, Sichuan Chansu may have a strong treatment effect on certain diseases, while Anhui Chansu may be more effective against other diseases). Whether and how the differences in quality of Chansu can cause the differences in medicine efficacy need to be further studied.

Point 2: Line 153 Figure1.The total ion chromatograms (TICs) of Chansu methanol extract in the positive ion mode. "Figure1" should add a space, such as Figure 1.

Response 2: Thanks for reviewer’s detailed suggestion. It has been modified

according to the reviewer’s suggestion. Line 159.

Point 3: Line 155 Figure2.The extracted ion chromatograms (XICs) of the main

compounds in the positive ion mode. "Figure2" should add a space, such as Figure 2.

Response 3: Thanks for reviewer’s kind suggestion. It has been modified according to the reviewer’s suggestion. Line 159. Line 161.

Point 4: Line 232: generated automatically by median method based onthe

chromatographic information. "onthe" should add a space. "median method" should be the median method.

Response 4: Thanks for reviewer’s kind suggestion. It has been modified according to the reviewer’s suggestion. Line 272.

Round 2

Reviewer 1 Report

This manuscript has improved its clarity, but I still have some minor suggestions as follows:

The proofreading of the text is critical as still many mistakes being found throughout. For examples,

L49: "......have a narrow window......."

L57: "of" in front of "among" should be deleted.

For the Introduction section:

LL73-85 should be selectively split and moved to either Materials and Methods or Discussion section.

(It is uncommon to describe those parts in the Introduction of a paper.)

L85: ".......We demonstrated......." (please delete "It is....."

LL88-89: Please delete the last sentence.

For the rest, this reviewer strongly encourages the authors to double check it carefully and also have it proofread by an English proficiency scientist of the same field of study. 

Author Response

Point 1: 

This manuscript has improved its clarity, but I still have some minor suggestions as follows:

The proofreading of the text is critical as still many mistakes being found throughout. For examples,

L49: "......have a narrow window......."

L57: "of" in front of "among" should be deleted.

Response 1:

Thank you for reviewing our manuscript during your busy schedule. Thank you very much for your kind suggestions to further improve the quality of manuscript. Thank you for your recognition and evaluation of this manuscript.

My colleagues and I have carefully proofread the text and corrected the mistakes in the manuscript.

L49: "......have a narrow window......."

It has been corrected.

L57: "of" in front of "among" should be deleted.

It has been deleted according to the reviewer’s suggestions.

In addition, we have corrected the mistakes in the manuscript using the “Track Changes” function in Microsoft Word.

Point 2: 

For the Introduction section:

LL73-85 should be selectively split and moved to either Materials and Methods or Discussion section.

(It is uncommon to describe those parts in the Introduction of a paper.)

Response 2:

Thanks for reviewer’s kind suggestion. It has been modified according to the reviewer’s suggestion.

Point 3:

L85: ".......We demonstrated......." (please delete "It is....."

Response 3:

Thanks for reviewer’s helpful suggestion. It has been modified according to the reviewer’s suggestion.

Point 4:

LL88-89: Please delete the last sentence.

Response 4:

Thanks for reviewer’s kind suggestion. It has been deleted according to the reviewer’s suggestion.

Point 5:

For the rest, this reviewer strongly encourages the authors to double check it carefully and also have it proofread by an English proficiency scientist of the same field of study. 

Response 5:

    Thanks for reviewer’s kind suggestion. My colleagues and I have checked the entire text carefully and corrected the mistakes in the text. At the same time, we also asked an English proficiency scientist in this field to further proofread the text. Finally, the manuscript was revised for proper English language, grammar, punctuation, spelling, and overall style by two of the highly qualified English-speaking scientists.

Reviewer 2 Report

Thank you for your extensive explanation which clarify some ambiguities in the manuscript. I recommend the publication of the manuscript in the present form. 

Author Response

Comments and Suggestions for Authors

Thank you for your extensive explanation which clarify some ambiguities in the manuscript. I recommend the publication of the manuscript in the present form. 

Response:

Firstly, thank you for reviewing our manuscript during your busy schedule. Secondly, thank you very much for your valuable suggestions to further improve the quality of manuscript. Finally, thank you for your recognition and evaluation of this manuscript.